

# Insulator phases of Bose-Fermi mixtures induced by intraspecies next-neighbor interactions

Felipe Gómez-Lozada[1], Roberto Franco[1] and Jereson Silva-Valencia[1,2]*

**1** Departamento de Física, Universidad Nacional de Colombia,
Bogotá D.C. 111321, Colombia.
**2** Department of Mechanical Engineering and Materials Science,
University of Pittsburgh, Pittsburgh, PA, USA.

* jsilvav@unal.edu.co

## Abstract

We study a one-dimensional mixture of two-color fermions and scalar bosons at the hard-core limit, focusing on the effect that the intraspecies next-neighbor interactions have on the zero-temperature ground state of the system for different fillings of each carrier. Exploring the problem's parameters, we observed that the nearest-neighbor interaction could favor or harm the well-known mixed Mott and spin-selective Mott insulators. We also found the emergence of three unusual insulating states with charge density wave (CDW) structures in which the orders of the carriers are out of phase between each other. For instance, the immiscible CDW appears only at half-filling bosonic density, whereas the mixed CDW state is characterized by equal densities of bosons and fermions. Finally, the spin-selective CDW couples the bosons and only one kind of fermions. Appropriate order parameters were proposed for each phase to obtain the critical parameters for the corresponding superfluid-insulator transition. Our results can inspire or contribute to understanding experiments in cold-atom setups with long-range interactions or recent reports involving quasiparticles in semiconductor heterostructures.

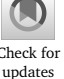

# 1  Introduction

The development of quantum simulators over the last few decades has allowed the control of many-body quantum systems' interactions and dynamics to a great extent [1, 2]. This paved the way to understanding the physics behind Bose–Fermi mixtures: composition of particles with both bosonic and fermionic statistics, which have been studied for decades with one of the first examples being the $^3$He-$^4$He combination [3–5]. Here the development of cold-atom simulators has been crucial [6–8], as it has been possible to produce an amalgam of degenerate mixtures of gases [9–26] which exhibit novel and fascinating phenomena such as Bose–Fermi superfluidity [27–30] or dual Mott insulators [31]. Likewise, recent experiments of dipolar excitons in 2D materials [32] present a promising alternative in the simulation of long-range interacting Bose–Fermi mixtures, since this type of interaction is inherent to these quasiparticles which enriches the physics of the system [33–39].

To perform a theoretical study of these mixtures, it is usual to describe fermions and bosons using the Hubbard model and coupling terms that account for interspecies interactions. Within this framework, there have been abundant numerical and analytical studies, which have resulted in the prediction of diverse kinds of carriers' configurations, such as Luttinger liquids, charge density waves (CDW), and Mott insulators (MI), among others [40–65]. Of particular interest for this research are the superfluid-insulator transitions, where it is well known that the insulator phases are characterized by commensurable relations between the fixed fermionic ($\rho_F$) and bosonic ($\rho_B$) densities. Specifically, for a two-color fermion and scalar boson mixture with repulsive interspecies interactions, there is always one mixed Mott insulator (MMI) characterized by the relation $\rho_B + \rho_F = n$ ($n$ being an integer) and a spin-selective Mott insulator (SSMI) that follows $\rho_B + \rho_F^{\uparrow,(\downarrow)} = n$, which indicates that for a population imbalance, this last insulator is divided in two [66–70]. It is important to note that flavor-selective Mott phases have been observed experimentally in SU(3) Fermi Hubbard models where the symmetry is explicitly broken by adding an inter-band hopping [71].

Long-ranged interactions have enriched the ground-state phase diagram of both bosonic and fermionic systems allowing diverse carrier configurations. Specifically, adding next-neighbor interactions to a bosonic system led to the emergence of exotic phases including supersolid, Haldane, and solitonic ones, and the charge density wave phase for integer and half-integer densities [72–78]. On the other hand, the one-dimensional phase diagram of fermions under short and long-ranged interactions is well-known for half-filling. However, it is controversial and little studied for other fillings [79–86]. At half-filling, a charge density wave phase emerges when the long-range interaction prevails over the on-site one, and a spin density wave when the opposite happens. In addition, other phases appear, such as a bond density wave when both interactions are of the same order, a phase separation for attractive interactions, and certain superconductor phases, singlet and triplet.

Motivated by the latter, we propose an exploration of the extended Bose–Fermi Hubbard model with two-color fermions and scalar bosons at the hardcore limit, including next-neighbor intraspecies interactions. Here we wish to discover what new phenomena can emerge from the interplay between long-range interactions and disparity of particle statistics. Moreover, the inclusion of spinful fermions allows the appearance of spin-selective phases, which are also of high interest [66–70], and the restriction in the form of the hardcore limit keeps a tractable Hilbert space for simulation while also allowing the extrapolation of its results to soft-core models, as shown in previous articles [67, 68]. There are only a few studies where these effects are considered for Bose–Fermi mixtures, with one example being an investigation with polarized fermions [87], so we believe the present study will significantly expand our understanding of this topic. Furthermore, the convergence of on-site and long-range interactions leads us to a multi-parameter model that could be essential for describing the physics of recent Bose-Fermi mixtures present in semiconductor heterostructures, where stripes and diverse bidimensional patterns have been observed [33–39].

In the following, we obtained a phase diagram for each type of long-ranged interaction, bosonic or fermionic, which summarizes the different insulator phases that can appear at a given filling (see Fig. 1). The latter contains the well-known mixed and spin-selective Mott insulators, as well as three unusual insulators that appear when taking into account the next-neighbor intraspecies interactions, all of which have charge density wave (CDW) orders for each species that are out of phase between each other. These correspond to the immiscible charge density wave (ICDW), a CDW phase with a period of two sites that only appears for half filling of the bosons $\rho_B = 1/2$, the mixed charge density wave (MCDW) in which both bosons and fermions have the same density, that is $\rho_B - \rho_F = 0$, and the CDW order is characterized by a wave vector proportional to said density, and the spin-selective charge density wave (SSCDW) that corresponds to an analogous phase to the latter where the bosons only couple to one of the fermionic spin components $\left(\rho_B - \rho_F^{\uparrow,(\downarrow)} = 0\right)$.

Therefore, the main objective of this article is to study the conditions under which these CDW insulators appear, as well as to expose the main properties that characterize their structure. To achieve this, in Sec. 2 we first present the extended Bose–Fermi Hubbard model to be used, along with a discussion of our numerical techniques, which are matrix product states (MPS) optimization algorithms for finding the ground state. The main results are found in Sec. 3, where we perform the analysis of Fig. 1(a) by choosing three representative fermionic densities given by $\rho_F = 2/5, 2/3, 4/5$, each of which shows a particular CDW insulator for nonzero fermionic next-neighbor interactions along with the principal characteristics from the phase diagram. Accordingly, in each subsection, we study the corresponding $\rho_B - \mu_B$ and $\mu_B - V_{FF}$ diagrams in conjunction with appropriate density profiles as a means of analyzing the properties of the different incompressible phases and their dependence on the fermionic next-neighbor interaction, with a particular focus on the ones of CDW type. In Sec. 4 we take a look at Fig. 1(b) and the role of bosonic next-neighbor interactions. Finally, in Sec. 5 we give some final remarks and future perspectives for research.

## 2 Model

A mixture of bosonic and fermionic atoms is commonly described by taking into account a local interaction term between bosons and fermions plus a Hubbard-type Hamiltonian for each
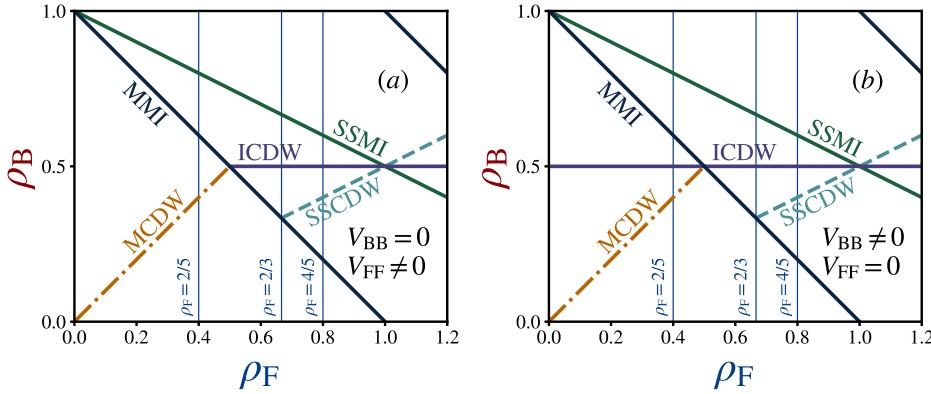

Figure 1: Phase diagram as a function of the bosonic $\rho_B$ and fermionic $\rho_F$ densities for a balanced ($\rho_F^\uparrow = \rho_F^\downarrow$) Bose–Fermi mixture with intraspecies next-neighbor interactions between only fermions (a) and bosons (b), respectively. Here we consider only rational values for both densities, guaranteeing that the number of carriers is always commensurate with the chain. Each non-vertical line corresponds to the density combinations necessary for the emergence of a particular insulator state, while the rest of the diagram shows a superfluid phase. The insulators correspond to the well-known mixed Mott insulator (MMI) at $\rho_B + \rho_F = 1$ (dark blue line) and the spin-selective Mott insulator (SSMI) at $\rho_B + \rho_F/2 = 1$ (dark green line), as well as the three phases found: the immiscible charge density wave (ICDW) at $\rho_B = 1/2$ (purple line), the mixed charge density wave (MCDW) at $\rho_B - \rho_F = 0$ (brown line) and the spin-selective charge density wave (SSCDW) at $\rho_B - \rho_F/2 = 0$ (light green line). The vertical lines signal special densities that are related to Fig. 2, 4, 6 and 11, here each time these vertical lines intersect with another non-vertical one an insulator phase appears in the corresponding phase diagram (for suitable values of the couplings). The graph only includes part of the range $\rho_F > 1$, since this region corresponds to a reflection of the $\rho_F < 1$ diagram thanks to the particle-hole symmetry of the Hamiltonian shown in Appendix A.

species, leading to the following expression:

$$\hat{\mathcal{H}} = \hat{\mathcal{H}}_B + \hat{\mathcal{H}}_F + \hat{\mathcal{H}}_{BF}, \tag{1}$$

$$\hat{\mathcal{H}}_B = -t_B \sum_{\langle j,l \rangle} \left( \hat{b}_j^\dagger \hat{b}_l + \text{H.c.} \right) + \frac{U_{BB}}{2} \sum_j \hat{n}_j^B \left( \hat{n}_j^B - 1 \right) + V_{BB} \sum_{\langle j,l \rangle} \hat{n}_j^B \hat{n}_l^B, \tag{2}$$

$$\hat{\mathcal{H}}_F = -t_F \sum_{\langle j,l \rangle, \sigma} \left( \hat{f}_{\sigma,j}^\dagger \hat{f}_{\sigma,l} + \text{H.c.} \right) + U_{FF} \sum_j \hat{n}_{\uparrow,j}^F \hat{n}_{\downarrow,j}^F + V_{FF} \sum_{\langle j,l \rangle} \hat{n}_j^F \hat{n}_l^F, \tag{3}$$

$$\hat{\mathcal{H}}_{BF} = U_{BF} \sum_j \hat{n}_j^B \hat{n}_j^F, \tag{4}$$

which is defined on a one-dimensional chain of length $L$ with open boundary conditions for numerical efficiency. In the above Hamiltonian $\hat{b}_j^\dagger, \hat{b}_j \left( \hat{f}_{\sigma,j}^\dagger, \hat{f}_{\sigma,j} \right)$ are the creator and annihilator boson (fermion with spin $\sigma = \uparrow, \downarrow$) operators at the j-th site with their corresponding number operators $\hat{n}_j^B = \hat{b}_j^\dagger \hat{b}_j$, $\hat{n}_{\sigma,j}^F = \hat{f}_{\sigma,j}^\dagger \hat{f}_{\sigma,j}$ and $\hat{n}_j^F = \hat{n}_{\uparrow,j}^F + \hat{n}_{\downarrow,j}^F$. The hopping integral for bosons (fermions) is $t_B$ ($t_F$), where $\langle j,l \rangle$ indicates next-neighbor pairs, and $U_{BB}$ ($U_{FF}$) is the intensity of the local boson-boson (fermion-fermion) interaction, while the magnitude of the local interspecies interaction is given by $U_{BF}$. Knowing that next-neighbor interactions are relevant, we consider the corresponding intraspecies terms for both fermions and bosons with strengths quantified by $V_{FF}$ and $V_{BB}$, respectively.

Additionally, the previous Hamiltonian admits a set of appropriate Abelian quantum numbers given by the boson number $N_B$ and the fermion number $N_F^\sigma$ with spin $\sigma = \uparrow, \downarrow$. From them we define useful quantities for the rest of the study, such as the total fermion number $N_F = N_F^\uparrow + N_F^\downarrow$, the global boson density $\rho_B = N_B/L$, and the global fermion densities $\rho_F^\sigma = N_F^\sigma/L$, and $\rho_F = N_F/L$. We also define $I = |\rho_F^\uparrow - \rho_F^\downarrow|/\rho_F$ as the imbalance of spin population, then we refer to the case of $I = 0$ as a balanced mixture.

Since the local Hilbert space for bosons is computationally intractable, we perform a cut in this local dimension by assuming the hardcore limit, which implies that there can be at most one boson per site. The latter set a local basis of dimension 8 given by all of the combinations $|n_B\rangle |n_{F\uparrow}\rangle |n_{F\downarrow}\rangle$ where each carrier number is either 0 or 1. Also, the hardcore limit is equivalent to assuming an infinite on-site boson interaction ($U_{BB} \to \infty$), and hence we do not consider this process for the rest of the paper. To further simplify our analysis, we suppose that $t_B = t_F = 1$, while also establishing our energy scale; therefore, all quantities are measured with respect to them. This last assumption is supported by the fact that these mixtures are usually constructed with isotopes of the same atom, for example with the ${}^6$Li-${}^7$Li [88] or ${}^{171}$Yb-${}^{174}$Yb [89] combinations.

To achieve insight into the physics of this model, we perform a ground-state energy search for a given set of particle fillings $E(N_B, N_F^\uparrow, N_F^\downarrow)$ and different interaction strengths, using a two-site density matrix renormalization group (DMRG) algorithm based on the MPS language [90] with the help of the TeNPy library [91]. We employ a sweep-dependent maximum bond dimension that starts at a given value between 500 and 1000 and increases every 10 sweeps by 100. The two-site DMRG algorithm truncates the Schmidt coefficients up to a maximum error of $10^{-14}$ or to the maximum bond dimension of the corresponding sweep; nevertheless, since the addition of next-neighbor interactions gives rise to a need for extra degrees of entanglement, the latter situation is the general case for our simulations. The stop criteria for the algorithm are determined by the energy and entropy errors, given by

$$\Delta E_n = \left| \frac{E_n - E_{n-1}}{\max(E_n, 1)} \right|, \tag{5}$$

$$\Delta S_n = \left| \frac{S_n - S_{n-1}}{S_n} \right|. \tag{6}$$

Here, $E_n$ and $S_n$ are the ground-state energy and the entanglement entropy at the middle of the lattice at sweep $n$. The denominator in $\Delta E_n$ is chosen to avoid numerical overflow if $E_n \approx 0$, which is not necessary in the case of entropy. In our program, we set maximum energy and entropy errors of $10^{-5}$ and $10^{-3}$, respectively, so when the wave function has errors above these bounds, the optimization stops, which proved to be enough to achieve convergence.

# 3 Fermionic next-neighbor interactions

The key findings of this work are shown in the phase diagrams from Fig. 1, where the different insulator phases that can appear at a given filling are summarized. Each diagram assumes that all densities attain rational values, which guarantees the commensurability between the chain and each component density. Moreover, since all density relations that define the possible configurations that a certain insulator can have are linear functions of $\rho_B$, $\rho_F^\uparrow$, and $\rho_F^\downarrow$ (see Sec. 1), each insulator is represented by a straight line in the phase diagram. This schematic also includes the results already known for the single component models: For $\rho_B = 0$ we have the band insulators at $\rho_F = 0$ and 2, and the Mott insulator at half-filling ($\rho_F = 1$), well known from the original Fermi-Hubbard model [92]. On the other hand, for $\rho_F = 0$ we find the analogous band insulator at $\rho_B = 0$ and the bosonic Mott insulator at unit-filling

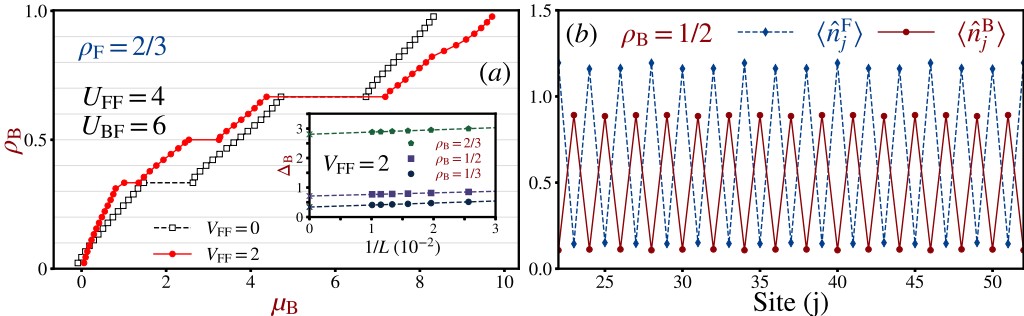

Figure 2: (a) Bosonic density $\rho_B$ vs. bosonic chemical potential $\mu_B$ at the thermodynamic limit for a balanced mixture with a fermionic density $\rho_F = 2/3$ and local interactions $U_{FF} = 4$, $U_{BF} = 6$. Here we compare the behavior with and without next-neighbor interaction between fermions, using $V_{FF} = 0, 2$. The inset shows the bosonic charge gap $\Delta_B$ as a function of the system size $L$ for each plateau found for $V_{FF} = 2$. In (b), we show the density profile for bosons $\langle \hat{n}_j^B \rangle$ (red circles) and fermions $\langle \hat{n}_j^F \rangle$ (blue diamonds) for $L = 75$ at the plateau with $\rho_B = 1/2$ found in (a). The points correspond to DMRG results, and the lines are visual guides.

($\rho_B = 1$) [74, 93–96]. In the case of $V_{BB} \neq 0$ (Fig. 1(b)) the CDW insulator is found for bosonic chains with next-neighbor interactions at $\rho_B = 1/2$, which has been found in previous works [50, 76, 77].

Now we focus our analysis on nonzero fermionic next-neighbor interactions only at three essential fermionic densities given by $\rho_F = 2/3, 2/5, 4/5$ at zero population imbalance $I = 0$, showing for each of them the emergence of an unknown insulator, along with a discussion on how the next-neighbor interactions between fermions affect the already well-known insulator phases. For these investigations, we fix the on-site interactions at $U_{FF} = 4$ and $U_{BF} = 6$, while we choose a next-neighbor interaction, which will be within the range $0 \leq V_{FF} \leq 6$. It has been shown in previous work that the presence of a next-neighbor interaction in the model leads to finite-size effects that perturb the ground-state energy if the number of sites is even [74], hence we restrict $L$ to be always odd. We also look at the imbalanced case $I \neq 0$ and how this affects the insulating phases of the model.

To detect the insulator phases, we measure the bosonic chemical potential

$$\mu_B(N_B, N_F^\uparrow, N_F^\downarrow) = E(N_B, N_F^\uparrow, N_F^\downarrow) - E(N_B - 1, N_F^\uparrow, N_F^\downarrow), \tag{7}$$

while varying the bosonic density from zero to one and looking for plateaus in the $\rho_B - \mu_B$ curve, which indicates the presence of an insulator, along with an extrapolation to the thermodynamic limit to determine the corresponding gap

$$\Delta_B(N_B, N_F^\uparrow, N_F^\downarrow) = \mu_B(N_B + 1, N_F^\uparrow, N_F^\downarrow) - \mu_B(N_B, N_F^\uparrow, N_F^\downarrow). \tag{8}$$

## 3.1 Two thirds fermionic filling ($\rho_F = 2/3$)

In Fig. 2(a), we show the bosonic density $\rho_B$ as a function of the bosonic chemical potential $\mu_B$ for $V_{FF} = 0$ (black open squares) and $V_{FF} = 2$ (red filled circles). For the turned-off next-neighbor interaction, we found a monotonous increase in $\mu_B$ as we continuously add bosons to the system in the majority of the graph, which corresponds to the presence of superfluid (SF) phases since they have nonzero compressibility. In particular, this behavior finds its exceptions at $\rho_B = 1/3, 2/3$ where the graph is discontinuous; we found incompressible phases. Each plateau in the $\rho_B - \mu_B$ curve represents the presence of an insulator state, where we denote

its charge gap $\Delta_B$ as the length of the plateau according to (8). At $\rho_B = 1/3$, we obtained a charge gap of $\Delta_B^{\rho_B=1/3} = 1.16$ corresponding to the well-known mixed Mott insulator (MMI) in which the interplay between commensurability of both bosons and fermions and their mutual interaction generate the presence of an incompressible phase. On the other hand, for $\rho_B = 2/3$, a plateau with a charge gap of $\Delta_B^{\rho_B=2/3} = 2.01$ appears, which we identify as a spin-selective Mott insulator (SSMI) that has an analog behavior as does the previous phase, but instead, the association is between bosons and only one type of fermion (spin up or down). These insulator phases are already known in the literature [31, 66–70] and follow the relations $\rho_B + \rho_F = 1$ for the MMI and $\rho_B + \rho_F/2 = 1$ for the SSMI in a balanced Bose–Fermi mixture.

After turning on the next-neighbor interaction, we found ourselves in a similar context, where most of the graph is in a SF phase. We still observe both previously mentioned insulators, but their charge gaps have changed. For the MMI, we found that the gap decreases to $\Delta_B^{\rho_B=1/3} = 0.34$, less than a third of its original value. A homogeneous distribution of carriers is expected in the absence of next-neighbor interaction, making it expensive to increase the number of particles, but allowing long-ranged coupling between fermions leads to the formation of composites between bosons and fermions, which have a less homogeneous distribution of charge along the lattice, diminishing the energy gap. On the other hand, the SSMI slightly increases its gap to $\Delta_B^{\rho_B=2/3} = 2.81$, showing that this phase benefits from the long-ranged fermion interaction. In this case, only half of the fermions are coupled to the bosons $\left(\rho_F^{\uparrow,(\downarrow)} = 1/3\right)$; however, the excess in the number of bosonic carriers makes the distribution of charge more homogeneous, making the effect of the next-neighbor interactions less dramatic.

On observing Fig. 2(a) for $V_{FF} = 2$, we notice that an additional incompressible phase at $\rho_B = 1/2$ emerges with a charge gap of $\Delta_B^{\rho_B=1/2} = 0.71$. To characterize this mysterious insulator, we show the corresponding bosonic and fermionic density profiles in Fig. 2(b). Since the fermions repel each other, the optimal configuration is to leave space between them, which produces the characteristic oscillating pattern from the CDW phase. Moreover, the bosons also interact with the fermions locally, hence they locate themselves between them, which generates another CDW order. Since the patterns are out of phase with each other, we call this insulator an immiscible CDW (ICDW). This curious insulator always appears at $\rho_B = 1/2$ (see Fig. 1(a)), since only for this bosonic density is the CDW structure stable for both species. That is, if there were more bosons, they could not fit between the fermions, and if there were fewer bosons, the fermions could flow freely through the spaces left behind by the missing bosons, removing the insulator condition. A similar bi-dimensional pattern was observed with neutral and charged dipolar excitons of GaAs bilayers recently [34], which allows us to confirm the relevance of the next-neighbor interactions in this Bose–Fermi system.

Let's analyze what happens when we vary the next-neighbor interaction. To do this, in Fig. 3(a) we show the evolution of the respective charge gaps from each phase as we increase the value of the next-neighbor interaction from $V_{FF} = 0$ to $V_{FF} = 4$. First, we observe that for the ICDW phase at $\rho_B = 1/2$ there exists a critical value $V_{FF}^{*\,\rho_B=1/2} \approx 0.7$ from which the insulator appears; to put it another way, for $V_{FF} \geq V_{FF}^{*\,\rho_B=1/2}$ we find a nonzero charge gap associated with this incompressible phase. Moreover, this charge gap increases with $V_{FF}$, since the stability of the CDW order improves for a larger next-neighbor interaction.

To characterize this transition, we introduce the following order parameter $\mathcal{O}^{ICDW}$ for the ICDW phase

$$\mathcal{O}^{ICDW} = \frac{-1}{L^2} \sum_{j,l}^{L} (-1)^{j+l} \left\langle \hat{n}_j^B - \rho_B \right\rangle \left\langle \hat{n}_l^F - \rho_F \right\rangle, \tag{9}$$

This signals the simultaneous CDW order of both species as the multiplication of the intraspecies CDW order parameters, with a minus sign representing the out-of-phase mutual oscillation, exhibited in Fig. 2(b). Moreover, we subtract the total density from each cor-

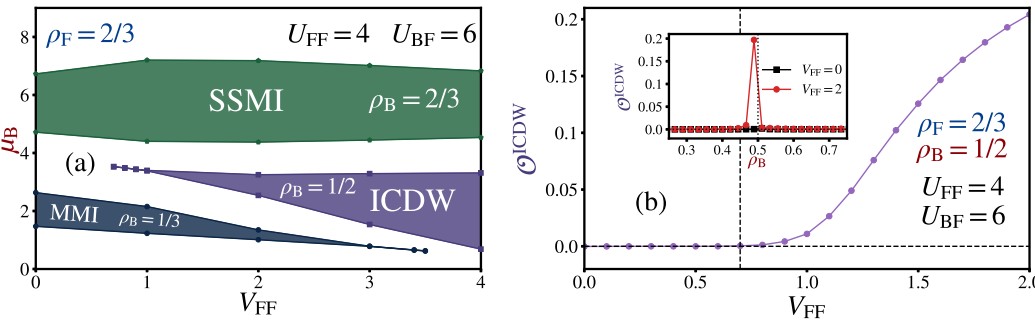

Figure 3: (a) Phase diagram of the bosonic chemical potential $\mu_B$ vs. next-neighbor fermion interaction magnitude $V_{FF}$ at the thermodynamic limit for a fermionic density $\rho_F = 2/3$ without spin imbalance and local interactions $U_{FF} = 4$ and $U_{BF} = 6$. Here we show the evolution of the respective charge gaps from each insulator phase shown in Fig. 2(a) which include the MMI at $\rho_B = 1/3$, the SSMI at $\rho_B = 2/3$ and the ICDW at $\rho_B = 1/2$. The points are extrapolations of the chemical potential to the thermodynamic limit. (b) Order parameter of the ICDW phase $\mathcal{O}^{ICDW}$ as a function of $V_{FF}$ in the thermodynamic limit for the same parameters as (a). The vertical dashed line denotes the critical value $V_{FF}^{*\,\rho_B=1/2}$ of the SF-ICDW transition obtained from the order parameter calculation. The inset shows the dependence of $\mathcal{O}^{ICDW}$ on $\rho_B$ for a chain of $L = 45$ and for $V_{FF} = 0$ (black squares) and $V_{FF} = 2$ (red circles). Here $\rho_B = 1/2$ is indicated as a dotted line. Horizontal dashed and dotted lines are added to highlight the zero value of the order parameter. Solid lines are visual guides.

responding density profile to remove the contributions of flat profiles. We show $\mathcal{O}^{ICDW}$ for $0 \leq V_{FF} \leq 2$ extrapolated to the thermodynamic limit in Fig. 3(b) where we can see the sudden change from zero to positive values after the critical point $V_{FF}^{*\,\rho_B=1/2} \approx 0.7$ (indicated by the vertical line) which agrees with the one obtained from the gap calculation. The dependence on the number of bosons is also shown in the inset of Fig. 3(b) as we vary $\rho_B$ and observe that the main nonzero contribution occurs close to half-filling for $V_{FF} = 2$ (red dots) while for $V_{FF} = 0$ (black squares) the order parameter is close to zero, as expected. Since we restrict ourselves to odd values of $L$ we observe a deviation of the maximum from $\rho_B = 1/2$, nevertheless by increasing the system size this finite-size effect disappears as the maximum gets closer to the expected value. Some remarks regarding the order of the transition can be found in Appendix B, a topic which falls outside the scope of this work.

On the other hand, for the MMI at $\rho_B = 1/3$, we see that the charge gap decreases monotonously until it vanishes at a critical value of $V_{FF}^{*\,\rho_B=1/3} \approx 3.5$, as anticipated from the formation of composites that destroy the crystal order mentioned beforehand. For the SSMI, even though we observed an initial increase in its charge gap, here we find that after a certain value, the gap starts to decrease monotonously. This behavior is persistent until the charge gap vanishes at a very high $V_{FF}$ value, not shown here. Although the excess of bosonic carriers and the fact that these tie only one kind of fermion together try to maintain a homogeneous distribution of charge in the system, large values of the next-neighbor interaction will force the emergence of compounds that diminish the ground-state energy and finally destroy the crystalline order.

It is worth noting that the ICDW phase does not appear for every fermionic density. Specifically, for $\rho_F < 1/2$, the system does not have enough fermions to build the sub-lattice for the CDW order. This is independent of the spin-population imbalance since the CDW order can be established independently of the number of fermions in each orientation. On the other hand, for $I \neq 0$ there is another limitation. Consider the case where $\rho_F^\uparrow$ or $\rho_F^\downarrow$ are larger than half

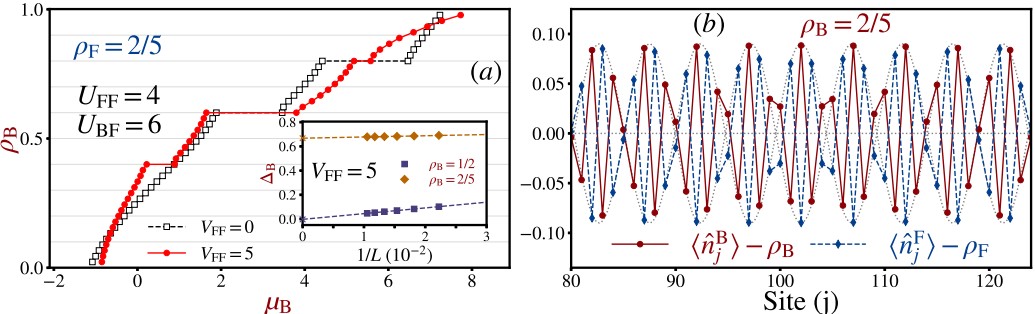

Figure 4: (a) Progress of the bosonic chemical potential of a balanced mixture as the number of bosons grows for a fermionic density $\rho_F = 2/5$ ($\rho_B - \mu_B$ curve). Here, we consider that $U_{FF} = 4$, $U_{BF} = 6$, and compare the behavior with and without next-neighbor interaction between fermions, using $V_{FF} = 0$, 5. The inset shows the charge gap $\Delta_B$ as a function of the system size $L$ for $V_{FF} = 5$ at $\rho_B = 2/5$, 1/2. In (b), we show the density profile for bosons $\langle \hat{n}_j^B \rangle$ (red circles) and fermions $\langle \hat{n}_j^F \rangle$ (blue diamonds) with respect to the corresponding densities $\rho_B$, $\rho_F$ for $L = 205$ at the plateau with $\rho_B = 2/5$ found in (a), along with dotted lines that only act as visual guides for the oscillating pattern. The points correspond to DMRG results, while the lines are visual guides.

filling; then all of the fermions with the corresponding spin do not fit in the CDW order, and hence it is not possible to construct the insulating phase. Both restrictions can be summarized with the mathematical condition that the ICDW phase only appears for $1/2 \leq \rho_F \leq 1/(1+I)$. The case $I = 1$, that is of spin-polarized fermions, has already been reported in the literature [87], and according to this study, there is only one possible density combination in which the insulator can appear, namely $\rho_F = \rho_B = 1/2$. This emphasizes that a mixture with a larger number of fermionic degrees of freedom admits a more general family of density combinations for the emergence of the ICDW state.

The presence of a non-zero bosonic gap shows the insulating characteristic of this mixture's component, so it is important to discuss the compressibility of the fermions in the system. Given that $\rho_B = 1/2$, the conditions for the emergence of the ICDW phase, that is $1/2 \leq \rho_F \leq 1/(1+I)$, delineate a region in the $\rho_F^\downarrow - \rho_F^\uparrow$ plane, inside of which we expect there to be no fermionic charge gap, since small deviations in the fermionic densities do not affect the conditions for the insulator to emerge. On the other hand, we do expect a fermionic gap at the boundaries, where the transition to the superfluid phases occurs, as is known to happen for half-filling ($\rho_B = \rho_F = 1/2$), where the state corresponds to the MMI [68], as an example. This is reminiscent of the supersolid phase, where crystalline and superfluid orders are present simultaneously [72,73]. Nevertheless, a further characterization falls outside the scope of this study.

## 3.2 Two fifths fermionic filling ($\rho_F = 2/5$)

From this point on, we will explore a lower fermionic density, $\rho_F = 2/5$, like that of Sec. 3.1; therefore, we plot the $\rho_B - \mu_B$ graph for $\rho_F = 2/5$ in Fig. 4(a), but in this case, the next-neighbor interaction is tuned between $V_{FF} = 0$ and 5. Without long-ranged interactions (black open squares), we obtain the already expected plateaus related to the MMI and SSMI at $\rho_B = 3/5$ and $\rho_B = 4/5$ respectively, with gaps $\Delta_B^{\rho_B=3/5} = 1.57$ and $\Delta_B^{\rho_B=4/5} = 2.05$. In the presence of next-neighbor interactions (red-filled circles), we can predict certain behaviors based on the previous discussion. Since we find ourselves in a region that fulfills $\rho_F < 1/2$, according

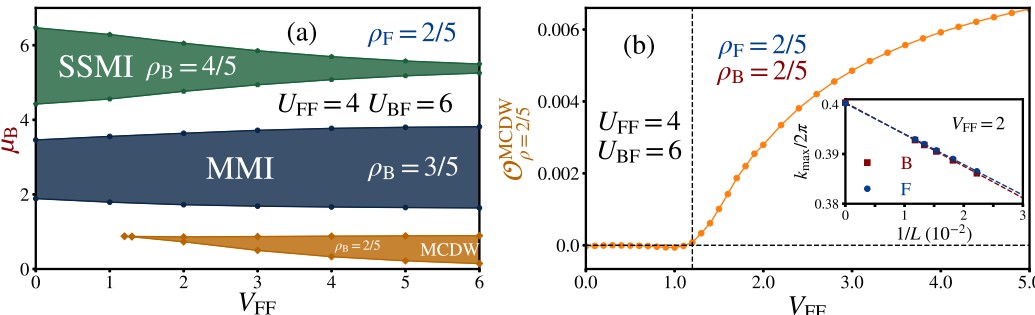

Figure 5: (a) Evolution of the insulating lobes with the next-neighbor fermion inter-action for a Bose–Fermi mixture with a fermionic density $\rho_F = 2/5$ and local inter-actions $U_{FF} = 4$ and $U_{BF} = 6$. The insulators are the ones shown in Fig. 4(a) which include the MMI at $\rho_B = 3/5$, the SSMI at $\rho_B = 4/5$ and the MCDW at $\rho_B = 2/5$. (b) MCDW order parameter $\mathcal{O}_\rho^{MCDW}$ at the corresponding density $\rho = 2/5$ as a function of $V_{FF}$ in the thermodynamic limit for the same parameters as (a). The vertical dashed line denotes the critical value $V_{FF}^{*\,\rho_B=2/5}$ after which $\mathcal{O}_\rho^{MCDW}$ differs from zero. A horizontal dashed line is added to highlight the zero value of the order parameter. Solid lines are visual guides. The inset shows the extrapolation to the thermodynamic limit of the wave vector of the largest contribution in the continuous Fourier trans-form $k_{max}/2\pi$ of the bosonic (red) and fermionic (blue) density profiles, for $V_{FF} = 2$. The dots correspond to extrapolations from DMRG results, while the lines are visual guides.

to Fig. 1(a) we do not observe a plateau associated with the ICDW, which is further empha-sized by considering the thermodynamic limit extrapolation shown in the inset of Fig. 4(a). Instead, the MMI and SSMI still emerge, but in this case the first one increases its charge gap to $\Delta_B^{\rho_B=3/5} = 2.15$, while the latter decreases its gap to $\Delta_B^{\rho_B=4/5} = 0.40$ when the long-ranged interaction is turned on, which is contrary to what happens in Fig. 2. For the MMI, since $\rho_F < 1/2$, the number of bosons is larger than the number of fermions, making it easy to insert bosons between fermions, which reduces the relevance of the next-neighbor interac-tion and makes it expensive to add bosons in an almost homogeneous distribution of carriers. On the other hand, the SSMI has a charge gap decrease due to the itinerant fermions, which modify the distribution of carriers, affecting the insulator. The above scenario remains as the fermionic next-neighbor interactions increase, as can be seen in Fig. 5(a).

Furthermore, because of the long-ranged interaction, an unusual incompressible phase emerges at $\rho_B = 2/5$, with a charge gap of $\Delta_B^{\rho_B=2/5} = 0.67$ and whose thermodynamic limit extrapolation is shown in the inset of Fig. 4(a). This particular insulator fulfills the relation $\rho_B - \rho_F = 0$, as shown in Fig. 1(a), which means, among other things, that a change in the number of fermions removes the insulator condition, hence a fermionic charge gap is expected, also given that this corresponds to the MMI for half-filling ($\rho_B = \rho_F = 1/2$) where such non-zero gap is known to appear [68]. For the sake of getting insight into this particular insulator, we take a look at the respective boson and fermion density profiles shown in Fig. 4(b), where we recognize two sets of CDW orders: a global one, which encloses both density profiles and a local CDW order analog to the ICDW insulator. Here the global order has a periodicity of five sites, with two particles of each species in every period, imposed by the mutual density $\rho_B = \rho_F = 2/5$. In comparison, the local order always has a period of two sites characteristic of a typical CDW phase. The latter behavior was checked for different fermionic density values from the respective phase line of Fig. 1(a); hence it corresponds to a key characteristic of the phase, and because of that, we will denote it a mixed CDW (MCDW).

When looking at the continuous Fourier transform of both the bosonic and fermionic profiles in the MCDW phase, we observe that the wave vector of the maximum contribution in the thermodynamic limit converges at $k = 2\pi\rho$ with $\rho = \rho_B = \rho_F = 2/5$ for each species, which can be seen for $V_{FF} = 2$ in the inset of Fig. 5(b). Based on this, we propose the following order parameter $\mathcal{O}_\rho^{MCDW}$ for the MCDW phase with $\rho = \rho_B = \rho_F$

$$\mathcal{O}_\rho^{MCDW} = \frac{-1}{L^2} \sum_{j,l}^{L} e^{i2\pi\rho(j+l)} \left\langle \hat{n}_j^B - \rho_B \right\rangle \left\langle \hat{n}_l^F - \rho_F \right\rangle , \qquad (10)$$

where we extend definition (9) to wave vectors different from $k = \pi/2$ and instead proportional to the mutual density as $k = 2\pi\rho$. Due to the inversion symmetry of the Hamiltonian (10), the above order parameter is always real in the thermodynamic limit, this is not true in general for an arbitrary finite lattice, hence for each simulation, we consider the absolute value of the order parameter, while the minus sign is assumed from the alternate order between species observed in the density profiles (Fig. 4(b)). In Fig. 5(b), we show $\mathcal{O}_{\rho=2/5}^{MCDW}$ in the MCDW phase for different $V_{FF}$ values, here we can see that the order parameter differs from zero after $V_{FF}^{*\,\rho_B=2/5} \approx 1.2$ which agrees with the gap opening of this phase as $V_{FF}$ increases shown in Fig. 5(a). In this case, the numerical calculation of $\Delta_B^{\rho_B=2/5}$ for small $V_{FF}$ shows to be numerically unstable, then the order parameter is best suited to obtain the critical value for the corresponding SF-MCDW transition. It is worth noting that the MCDW lobe grows as the next-neighbor interaction increases, as can be seen in Fig. 5(a). Analog to the ICDW analysis, we provide some insight into the order of the transition in Appendix B.

There is an additional finite-size effect associated with this phase. For a finite chain of size $L$ the system exhibits the properties of the MCDW phase at density $\rho$ for a reduced number of bosons given by $N_B' = \rho L - 1$. This was taken into account for the calculation of the corresponding bosonic charge gaps, density profiles, and order parameters from Fig. 4 and Fig. 5. For the thermodynamic limit extrapolation, the latter effect vanishes as $N_B'/L \approx \rho + \mathcal{O}(1/L)$. In particular, to obtain the most characteristic order parameter we first calculate $\mathcal{O}_{\rho'}^{MCDW}$ for a range of reference densities $\rho - 1/L \leq \rho' \leq \rho$ and use the value of maximum magnitude from this set as the finite-size value for the subsequent extrapolations to the thermodynamic limit.

The emergence of the MCDW phase requires nonzero next-neighbor interactions and a balance of the carrier densities (bosons and fermions), which suggest the formation of Bose–Fermi composites to establish this peculiar insulator state. Another interesting fact is that this incompressible phase can only appear for $\rho_B + \rho_F \leq 1$, meaning that all fermions must cooperate to form this unique insulator. The limit case when $\rho_B + \rho_F = 1$ corresponds to the mutual half-filling situation $\rho_B = \rho_F = 1/2$ where both orders attain a two-site periodicity, turning into the ICDW as $\mathcal{O}_{\rho=1/2}^{MCDW} = \mathcal{O}^{ICDW}$ from (9) and (10). The large unit cell of this CDW insulator and the fact that the corresponding densities are in general rational (in contrast to the MMI and SSMI, where the sum of two densities is an integer, and the ICDW, where the unit cell size is two, as expected from a usual CDW phase) shows similarity to ground states found in frustrated magnetism models characterized by rational magnetization plateaus [97–99]. In this sense, the high nearest-neighbor interaction could be interpreted as a source of frustration that will reduce the effective kinetic energy of the carriers, thus enhancing the stability of the non-trivial CDW insulator (see Fig. 5(a)). While this topic may deviate from the primary focus, it offers a promising avenue for future research on the characterization of this insulator.

## 3.3 Four fifths fermionic filling ($\rho_F = 4/5$)

As can be seen in Fig. 1(a), there is an unknown insulator state that has not been discussed yet; therefore, we fix the fermionic density at $\rho_F = 4/5$ and calculate the bosonic chemical

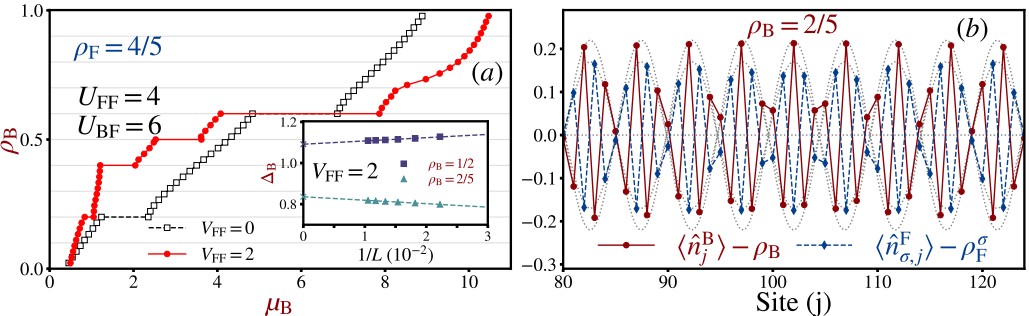

Figure 6: Bosonic density $\rho_B$ vs. bosonic chemical potential $\mu_B$ at the thermodynamic limit for a fermionic density $\rho_F = 4/5$. The local repulsion couplings are $U_{FF} = 4$, and $U_{BF} = 6$, and we compare the behavior with and without next-neighbor interaction between fermions using $V_{FF} = 0, 2$. The inset shows the charge gap $\Delta_B$ as a function of the inverse of the system size $L$ for $V_{FF} = 2$ at $\rho_B = 2/5, 1/2$. In (b), we show the density profile for bosons $\langle \hat{n}_j^B \rangle$ (red circles) and one color of fermions $\langle \hat{n}_{\sigma,j}^F \rangle$, that is either $\sigma = \uparrow$ or $\downarrow$, (blue diamonds) with respect to the corresponding densities $\rho_B, \rho_F^\sigma$ for $L = 205$ at the plateau with $\rho_B = 2/5$ found in (a), along with dotted lines that only act as visual guides of the oscillating pattern. The points correspond to DMRG results and the lines are visual guides.

potential as the number of bosons grows, which is displayed in Fig. 6(a). Also, we follow the evolution of the respective charge gaps from each plateau as we vary the next-neighbor interaction according to $0 \leq V_{FF} \leq 4$ in Fig. 7(a). In the case of null next-neighbor interaction between fermions, we find the expected MMI and SSMI plateaus at $\rho_B = 1/5, 3/5$ with charge gaps of $\Delta_B^{\rho_B=1/5} = 1.12$ and $\Delta_B^{\rho_B=3/5} = 2.02$. By including the long-ranged interaction, the ICDW insulator predicted in Sec. 3.1 emerges with a nonzero charge gap for next-neighbor interactions higher than the critical value $V_{FF}^{*\rho_B=1/2} \approx 0.5$, calculated using the ICDW order parameter (9) and corroborated by the gap calculations (see Fig. 7(a)). Then, as we increase $V_{FF}$ we observe that the charge gap from the ICDW insulator also increases monotonously, with the specific value of $\Delta_B^{\rho_B=1/2} = 1.09$ for $V_{FF} = 2$ corresponding to Fig. 6(a). On the other hand, in the presence of nonzero next-neighbor interactions, the MMI decreases its charge gap to $\Delta_B^{\rho_B=1/5} = 0.21$ for $V_{FF} = 2$ as a consequence of the formation of composites between fermions and bosons. The latter continues as we increase $V_{FF}$ until the corresponding plateau vanishes at $V_{FF}^{*\rho_B=1/5} \approx 2.7$ (see Fig. 7(a)). Also, we note that the charge gap of the SSMI state increases to $\Delta_B^{\rho_B=3/5} = 3.75$ for $V_{FF} = 2$, but as the next-neighbor interaction grows further, the gap starts to decrease, which is expected, since the free fermions of the spin-selective phase can disturb more the insulating structure with higher $V_{FF}$ values, hence reducing its stability. All of these scenarios are akin to the results found in Sec. 3.1 for the three corresponding insulators, since between $\rho_F = 2/3$ and $\rho_F = 4/5$ there are no particular changes in their behavior.

Nevertheless, we do find a peculiar plateau in Fig. 6(a), not seen in previous sections, when we turn on the next-neighbor interaction, which is located at $\rho_B = 2/5$ and has a charge gap of $\Delta_B^{\rho_B=2/5} = 0.83$. This particular incompressible phase appears when the density of the bosons equals half the density of the fermions; that is, $\rho_B - \rho_F/2 = 0$. As a means to shed light on its nature, in Fig. 6(b) we show its corresponding boson and fermion density profiles, where we can see similar behavior to the MCDW phase of Fig. 4(b) only with a difference in the oscillation amplitudes of each species and the charge of each period, which in this case is of two bosons and four fermions. In this case, the synchronization is between bosons and only half of the fermion population, we corroborate this by looking at the wave vector of the

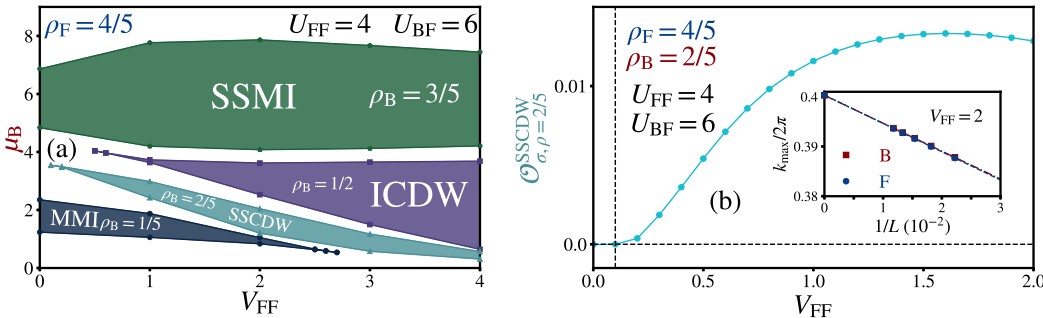

Figure 7: (a) Phase diagram of the bosonic chemical potential $\mu_B$ vs next-neighbor fermion interaction magnitude $V_{FF}$ at the thermodynamic limit for a fermionic density $\rho_F = 4/5$ without spin imbalance and local interactions $U_{FF} = 4$ and $U_{BF} = 6$. The colored regions exhibit the evolution of the insulator states shown in Fig. 6(a), which are surrounded by SF regions and include the MMI at $\rho_B = 1/5$, the SSMI at $\rho_B = 3/5$, the ICDW at $\rho_B = 1/2$ and the SSCDW at $\rho_B = 2/5$. (b) SSCDW order parameter $\mathcal{O}_{\sigma,\rho}^{SSCDW}$ at the corresponding density $\rho = 2/5$ for a balanced mixture as a function of $V_{FF}$ in the thermodynamic limit for the same parameters as (a). The vertical dashed line denotes the critical value $V_{FF}^{*,\rho_B=2/5}$ after which $\mathcal{O}_{\rho}^{SSCDW}$ differs from zero. A horizontal dashed line is added to highlight the zero value of the order parameter. The inset shows the extrapolation to the thermodynamic limit of the wave vector of the largest contribution in the continuous Fourier transform $k_{max}/2\pi$ of the bosonic (red) and fermionic (blue) density profiles, for $V_{FF} = 2$. The critical value $V_{FF}^{*,\rho_B=1/2}$ for the ICDW phase is calculated using the corresponding order parameter (9). The dots correspond to extrapolations from DMRG results, while the lines are visual guides.

highest contribution in each species profile, which extrapolated to the thermodynamic limit at $V_{FF} = 2$ tends to $k = 2\pi\rho$ with $\rho = \rho_B = \rho_F/2 = 2/5$ (see inset of Fig. 7(b)). This scenario is analogous to what happened with the MMI and SSMI phases in a balanced mixture [67], then we denote this insulator as a spin-selective CDW (SSCDW), in the following we will show that it has the properties of a spin-selective insulator.

Next, we explore the behavior of the system at the given fermionic density as we consider nonzero spin-population imbalance. Therefore, in Fig. 8(a) we exhibit the $\rho_B - \mu_B$ graph for $\rho_F = 4/5$ with a next-neighbor interaction of $V_{FF} = 1$ and different cases of spin-imbalance given by $I = 0$ and $I = 1/6$. Without spin imbalance (black open squares), we observe the expected plateaus corresponding to the MMI with gap $\Delta_B^{\rho_B=1/5} = 0.82$, the SSMI with gap $\Delta_B^{\rho_B=3/5} = 3.57$, and the SSCDW with a gap of $\Delta_B^{\rho_B=2/5} = 0.56$. We do not clearly see the ICDW plateau, not because there is none, but because its charge gap has a small value of $\Delta_B^{\rho_B=1/2} = 0.09$, which is difficult to note at this scale. After increasing the spin-population imbalance to $I = 1/6$, we find that the charge gap of the MMI barely changes to $\Delta_B^{\rho_B=1/5} = 0.86$, while the SSMI splits into two plateaus at $\rho_B = 8/15, 2/3$ with corresponding charge gaps of $\Delta_B^{\rho_B=8/15} = 2.00$ and $\Delta_B^{\rho_B=2/3} = 1.97$. On top of that, we notice that the plateau associated with the SSCDW phase at $\rho_B = 2/5$ also splits into two incompressible phases at $\rho_B = 1/3$, $7/15$ with charge gaps of $\Delta_B^{\rho_B=1/3} = 0.11$ and $\Delta_B^{\rho_B=7/15} = 0.54$, respectively, which is further emphasized by the inset in Fig. 8(a).

We show the corresponding density profiles for the insulator at $\rho_B = 7/15$ in Fig. 8(b), here we observe the characteristic CDW coupling between spin-up fermions and bosons with seven particles per period of each coupled component, indicating that it also corresponds to

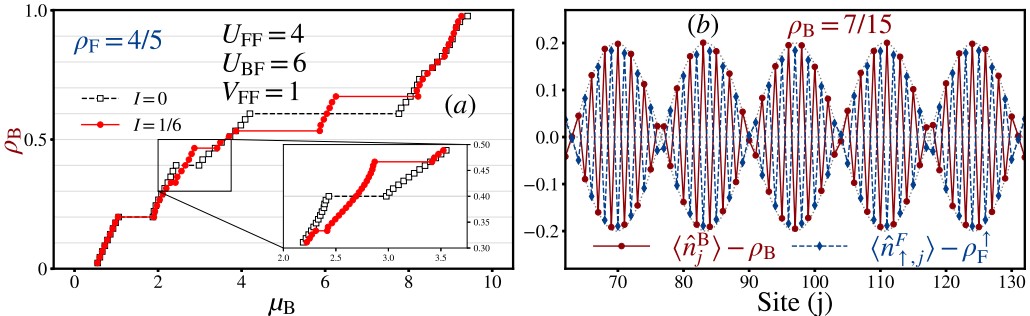

Figure 8: Bosonic density $\rho_B$ vs. bosonic chemical potential $\mu_B$ at the thermodynamic limit for a fermionic density $\rho_F = 4/5$, local interactions $U_{FF} = 4$, $U_{BF} = 6$, and next-neighbor interaction $V_{FF} = 1$. Here we compare the behavior with and without spin imbalance using $I = 0$, $1/6$. The inset shows a close-up of the region where the SSCDW splits because of the nonzero spin-population imbalance. In (b) we show the density profile for bosons $\langle \hat{n}_j^B \rangle$ (red circles) and spin-up fermions $\langle \hat{n}_{\uparrow,j}^F \rangle$ (blue diamonds) with respect to the corresponding densities $\rho_B$, $\rho_F^\uparrow$ for $L = 205$ at the plateau with $\rho_B = 7/15$, found in (a), along with corresponding dotted lines that only act as visual guides. The points correspond to DMRG results, while the lines are visual guides.

a SSCDW phase. The insulator at $\rho_B = 1/3$ also follows an analogous behavior with the corresponding periodicity, not shown here. Moreover, by looking at the bosonic correlation function $\langle \hat{b}_x^\dagger \hat{b}_{x+y} \rangle$[1] in Fig. 9(a) for the imbalanced case from Fig. 8(a) at $\rho_B = 2/5$ we see a potential decay indicating its SF nature until an edge effect appears as a sudden decrease in the correlation function, while the SSCDW plateaus at $\rho_B = 1/3$, and $7/15$ possess an exponential decay, characteristic of an insulator state, hence the SSCDW states are separated by a superfluid phase.

Now we focus on one of the splitted SSCDW insulators, specifically $\rho_B = 1/3$, and show that the uncoupled fermionic spin component is not insulating. For this we fix $\rho_B = \rho_F^\downarrow = 1/3$ and study the dependence of $\rho_F^\uparrow$ on its corresponding chemical potential $\mu_F^\uparrow$, which we calculate in an analog way as (7). This graph, shown in Fig. 9(b), exhibits the MMI and SSMI phases at $\rho_F^\uparrow = 1/3$, $2/3$, respectively with corresponding gaps $\Delta_{F,\uparrow}^{\rho_F^\uparrow = 1/3} = 0.93$ and $\Delta_{F,\uparrow}^{\rho_F^\uparrow = 2/3} = 2.12$, nevertheless for $\rho_F^\uparrow = 7/15$, density at which the bosonic charge gap is nonzero (see Fig. 8(a)), the spin-up fermionic gap is closed in the thermodynamic limit, emphasized by the inset of Fig. 9(a). This indicates that the spin-up fermions are in a gapless phase inside this SSCDW insulator.

After establishing the spin-selective character of the SSCDW phase, we propose a corresponding order parameter $\mathcal{O}_{\sigma,\rho}^{SSCDW}$ as an extension of (10)

$$\mathcal{O}_{\sigma,\rho}^{SSCDW} = \frac{-1}{L^2} \sum_{j,l}^{L} e^{i2\pi\rho(j+l)} \left\langle \hat{n}_j^B - \rho_B \right\rangle \left\langle \hat{n}_{\sigma,l}^F - \rho_F^\sigma \right\rangle, \tag{11}$$

where we specify the spin as $\sigma = \uparrow, \downarrow$.[2] In Fig. 7(b), we show $\mathcal{O}_{\sigma,\rho=2/5}^{SSCDW}$ for a range of $V_{FF}$ values

---

[1]The correlation functions $\langle \hat{b}_x^\dagger \hat{b}_{x+y} \rangle$ are calculated by averaging over $x$ to remove finite size effects due to the CDW oscillations in the system.

[2]The SSCDW phase exhibits the same finite-size effect as the MCDW phase where one has to measure its properties with one boson less than the expected, then we use the same procedure from Sec. 3.2 for the calculation of the bosonic gaps, density profiles, and order parameters.

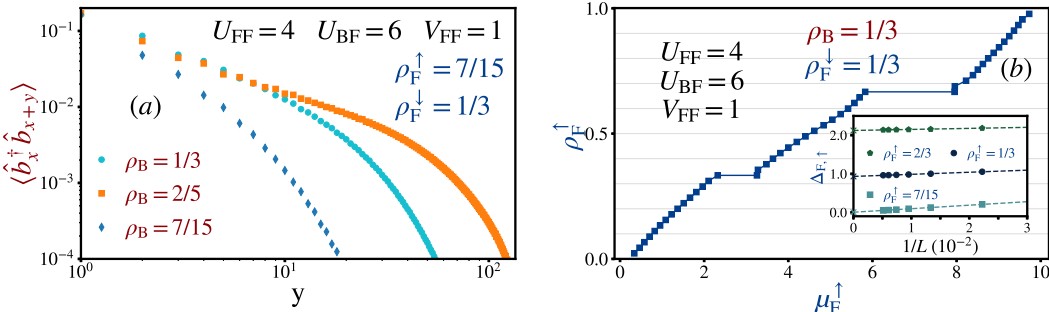

Figure 9: (a) Bosonic correlation function $\langle \hat{b}_x^\dagger \hat{b}_{x+y} \rangle$ averaged over $x$ as a function of $y$ for a lattice of size $L = 135$ at the fermionic densities $\rho_F^\uparrow = 7/15$, $\rho_F^\downarrow = 1/3$ and different bosonic densities $\rho_B = 1/3$ (light blue), $2/5$ (orange) and $7/15$ (dark blue). We use a log-log scale to emphasize the superfluid or insulating behavior of the correlation functions and consider the interaction parameters $U_{FF} = 4$, $U_{BF} = 6$ and $V_{FF} = 1$. (b) Spin-up fermionic density $\rho_F^\uparrow$ vs the corresponding fermionic chemical potential $\mu_F^\uparrow$ at the thermodynamic limit with interactions $U_{FF} = 4$, $U_{BF} = 6$, $V_{FF} = 1$ and the rest of the densities given by $\rho_B = 1/3$ and $\rho_F^\downarrow = 1/3$. The inset shows the extrapolations of the fermionic charge gap $\Delta_F^\uparrow$ for $\rho_F^\uparrow = 1/3$, $7/15$ and $2/3$. The lines are visual guides, while the points correspond to DMRG results.

with the parameters of the balanced case from Fig. 7(a) for which we use either $\sigma = \uparrow$ or $\downarrow$ and $\rho_F^\sigma = \rho_F/2$. We note that after the critical value $V_{FF}^{*\rho_B=2/5} \approx 0.1$ both the order parameter and the corresponding gap (Fig. 7(a)) increase from zero. For an imbalanced case, we checked that the definition (11) also works as an order parameter using the corresponding coupled fermionic component. Inside the balanced SSCDW phase, half of the fermions are uncoupled from the insulator structure, this decreases its stability with higher $V_{FF}$ as seen in a reduction of both the gap and order parameter in Fig. 7. For large enough $V_{FF}$, we expect this behavior to break the insulator and transition the ground state to a SF phase. Even then, the SSCDW phase appears only for $\rho_B + \rho_F \geq 1$ since in this region the chain gets saturated with all of the carriers, which forces the coupling to only contain one type of fermion and not both, in contrast to what happens with the MCDW. This condition is also independent of the spin-population imbalance since it only deals with the complete fermionic and bosonic densities.

## 3.4 Phase diagram for nonzero spin-population imbalance

According to the previous sections, there are two spin-selective phases in the present long-ranged mixture model, so it is of great value to analyze how the phase diagram from Fig. 1(a) changes as we increase the spin-population imbalance. This is depicted in Fig. 10, where we exhibit the phase diagram for $I = 0$, $1/6$, $1/2$ and $1$, using the corresponding density relations for non-zero spin-population imbalance found in previous sections. Here, we observe that the phase space configuration of the MMI (dark blue) and MCDW (brown) insulators is invariant under the change of the spin-population imbalance, which further emphasizes its spin-independent nature. On the other hand, as we increase $I$, we can see the splitting of the spin-selective phases. In the case of the SSMI (dark green), the separation grows until the spin-up insulator follows $\rho_B = 1$ while the other one matches the MMI line. For the SSCDW (light green), there is an additional constraint, $\rho_B + \rho_F \geq 1$; hence as the imbalance increases, the spin-down phase shortens until it vanishes at $I = 1$, while the spin-up phase follows the relation $\rho_B - \rho_F = 0$. In the case of the ICDW (purple line), the condition $1/2 \leq \rho_F \leq 1/(1+I)$ limits its phase space line as $I$ grows until it converges at the point $\rho_B = \rho_F = 1/2$. Hence, for

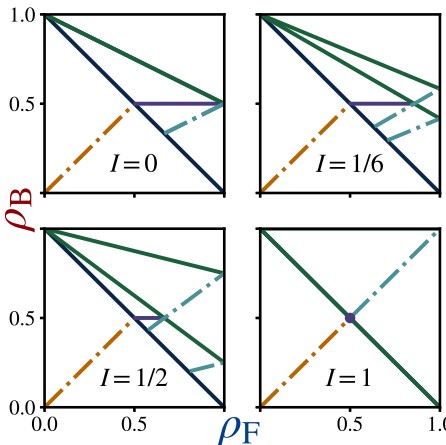

Figure 10: Phase diagram of the possible insulator phases that can appear in a Bose–Fermi mixture with next-neighbor fermionic interactions for different values of the spin-population imbalance $I = 0, 1/6, 1/2, 1$. The phases correspond to the MMI (dark blue line), MCDW (brown line), SSMI (dark green line), SSCDW (light green line), and ICDW (purple line and dot).

polarized fermions at half-filling the three different insulators found in this study follow the single condition $\rho_B - \rho_F = 0$. This emphasizes how a study with spinor fermions can show different emergent behaviors that can be elusive in an investigation with only scalar particles.

# 4 Bosonic next-neighbor interactions

Up to this point, we have only considered nonzero next-neighbor fermionic interactions, then a good question is what happens when the long-ranged interaction is between the bosons only. In order to answer this, we plot in Fig. 11 the $\rho_B - \mu_B$ graph for $\rho_F = 2/5, 2/3, 4/5$, keeping the local interactions constant at $U_{FF} = 4$ and $U_{BF} = 6$ while turning on the next-neighbor bosonic interaction $V_{BB}$ for values in the range $0 \leq V_{BB} \leq 6$. For $\rho_F = 2/3$ (Fig. 11(a)) and $V_{BB} = 2$ we find once again three plateaus corresponding to the MMI at $\rho_B = 1/3$, the SSMI at $\rho_B = 2/3$ and the ICDW at $\rho_B = 1/2$ with charge gaps $\Delta_B^{\rho_B=1/3} = 1.27$, $\Delta_B^{\rho_B=2/3} = 2.51$ and $\Delta_B^{\rho_B=1/2} = 1.18$, respectively. In this case, the main difference concerning $V_{FF} \neq 0$ is that the plateaus of the MMI increase slightly, instead of decreasing as it happens in Fig. 4(a), which shows that the charge gap has a dependence on the associated particle statistics of the long-ranged interactions. Apart from that, for $\frac{1}{2} \leq \rho_F \leq \frac{2}{3}$ we found that just as Fig. 11(a), the phase diagram for fermionic next-neighbor interactions is the same for bosonic next-neighbor interactions (See Fig. 1).

On the other hand, at $\rho_F = 2/5$ (Fig. 11(b)) we do observe an additional incompressible phase besides the ones in Fig. 4(a) when we turn on the long-ranged interaction to $V_{BB} = 6$. This new plateau is located at $\rho_B = 1/2$ and corresponds to the ICDW with charge gap $\Delta_B^{\rho_B=1/2} = 6.64$. Here, since the bosons possess long-ranged interactions they can form the CDW structure even in the absence of fermions, then in this case the ICDW appears for any fermionic filling, as can be seen in Fig. 1(b). Apart from this insulator, we still find the MMI and SSMI at $\rho_B = 3/5$ and $\rho_B = 4/5$ with gaps $\Delta_B^{\rho_B=3/5} = 2.09$ and $\Delta_B^{\rho_B=4/5} = 2.31$, respectively. For this instance, both of them increase their stability with higher $V_{BB}$. In Sec. 3.2

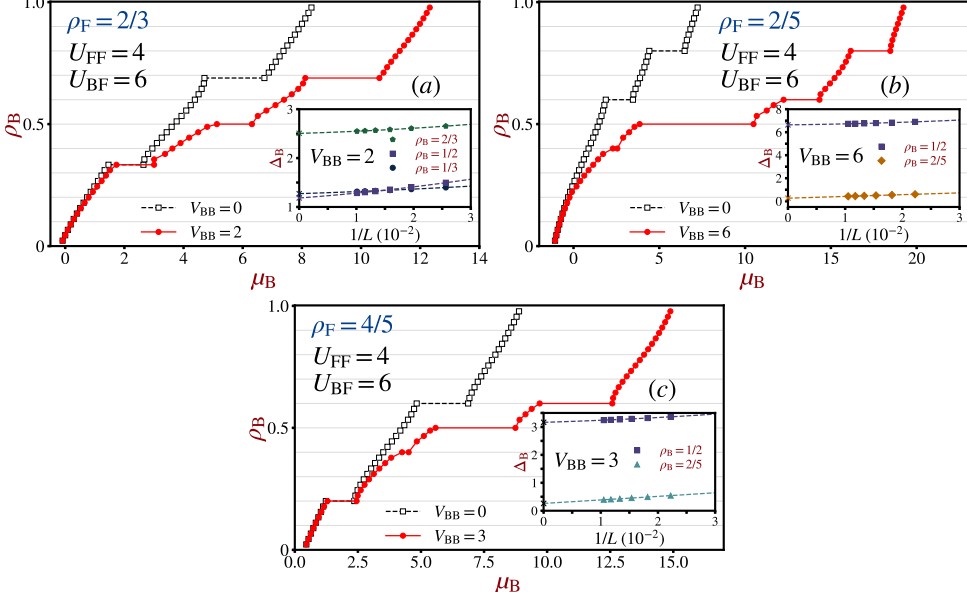

Figure 11: Bosonic density $\rho_B$ vs. bosonic chemical potential $\mu_B$ at the thermodynamic limit for fermionic densities (a) $\rho_F = 2/3$, (b) $\rho_F = 2/5$ and (c) $\rho_F = 4/5$. The local repulsion couplings are $U_{FF} = 4$, and $U_{BF} = 6$, and we compare the behavior with and without next-neighbor interaction between bosons using $V_{BB} = 0, 2, 3, 6$. Each inset shows the charge gap $\Delta_B$ as a function of the inverse of the system size $L$ for the corresponding nonzero $V_{BB}$ at characteristic bosonic densities $\rho_B$. The points correspond to DMRG results and the lines are visual guides.

for $V_{FF} \neq 0$ we found that the long-ranged interactions reduced drastically the spin-selective phase gap since the free fermions are enhanced by this interaction, and hence can disturb the insulating structure. In this case, only the bosons have long-ranged interactions, and because of that the stability of the phase is not damped. The MCDW phase still appears in the phase diagram but with a tiny charge gap of $\Delta_B^{\rho_B=2/5} = 0.26$, considering the high value of the bosonic interaction in this case, this suggests that the MCDW structure is more favorable for $V_{FF} \neq 0$.

Finally, in Fig. 11(c) we show the case for $\rho_F = 4/5$, which has analogous results as the previous two fermionic densities. The ICDW appears with a charge gap of $\Delta_B^{\rho_B=1/2} = 3.16$, as expected from Fig. 1(b). For the MMI and SSMI, their charge gaps increase to $\Delta_B^{\rho_B=1/5} = 1.16$ and $\Delta_B^{\rho_B=3/5} = 2.88$, respectively, the same way as it happened for Fig. 11(a). On the contrary, the SSCDW arises with a small charge gap of $\Delta_B^{\rho_B=2/5} = 0.26$ similar to the MCDW in Fig. 11(b).

In summary, the most relevant change to the phase diagram when we consider next-neighbor interactions between bosons is the presence of the ICDW phase for all values of the fermionic density $\rho_F$, which is depicted in Fig. 1.

## 5 Conclusions

In the present study, we have introduced three CDW insulators that emerge after taking into account next-neighbor intraspecies interactions for a Bose–Fermi mixture model of two-color fermions and scalar bosons at the hardcore limit. We used an MPS-based DMRG method to obtain the ground state for given carrier densities and interaction parameters and subsequently constructed plots of the bosonic density against the bosonic chemical potential to determine

the presence of the corresponding incompressible phases. We proposed order parameters, as well as constructed multiple density profiles and phase diagrams, varying the next-neighbor interaction between fermions or bosons to characterize the properties of the cited insulators.

The effect of the long-ranged interactions on the well-known mixed Mott insulator (MMI) [31,66] and the spin-selective Mott insulator (SSMI) [67–70] was studied, noting that those insulators tend to disappear or remain stable, depending on whether the value of the fermionic density is greater or less than $\rho_F = 1/2$ and the kind of species under long-ranged interactions. For instance, we observed that for $\rho_F < 1/2$ and next-neighbor interactions between fermions, the MMI (SSMI) phase remains stable (tends to disappear) when the long-ranged coupling grows, while the opposite happens for $\rho_F > 1/2$. The above scenario is exchanged if we consider long-ranged interaction between bosons instead of fermions.

An immiscible charge density wave (ICDW) phase emerges at $\rho_B = 1/2$, which has a CDW order for both fermions and bosons that are completely out of phase. It was established that this insulator only appears for $1/2 \leq \rho_F \leq 1/(1+I)$, which is associated with the system having just enough fermions to create the characteristic dimerized lattice from the CDW order. The gap from this incompressible phase grows with increasing next-neighbor fermion interaction since this long-ranged repulsion provides stability to the CDW structure. The presence of the phase was clarified by employing an order parameter constructed from a product of the traditional CDW order parameter for each species, with a minus sign that accounts for the out-of-phase oscillation. This quantity agrees with the gap opening and gives a stronger argument for the determination of the critical point. Exchanging fermions with bosons for the next-neighbor interaction removes the left border of this phase, allowing this phase to occur at any fermionic density.

On the other hand, another non-trivial insulator appears for the condition $\rho_B - \rho_F = 0$, where both species have CDW orders with a characteristic wave vector proportional to the density of both species, which we denote as mixed charge density wave (MCDW). We characterize its critical transition using an order parameter analog to the ICDW one where we change the wave vector to the corresponding one for the MCDW phase, which we corroborate with the bosonic gap calculations. This phase only appears for $\rho_B + \rho_F \leq 1$ and for a fermionic density of $\rho_F = 2/5$ it has a unit cell of length five sites that contains two particles of each species. This phase is favored, and its bosonic gap increases with the next-neighbor interaction between fermions or bosons.

A total number of carriers greater than the lattice size ($\rho_B + \rho_F \geq 1$) and interactions open the possibility to spin-selective states, a fact that was corroborated by the emergence of the spin-selective charge density wave (SSCDW) insulator when turning on the long-ranged interaction between fermions or bosons. In this state, one kind of fermion is itinerant, while the other couples with the bosons to establish an insulator state that fulfills $\rho_B - \rho_F^{\uparrow,(\downarrow)} = 0$ and exhibits interleaved CDW profiles with a global pattern. We showed explicitly the spin-selective character of this state by breaking the SU(2) symmetry, i.e., under a spin-population imbalance, this state splits into two SSCDW insulators, which are separated by a superfluid phase. Finally, a spin-selective order parameter was proposed and used to identify the superfluid-SSCDW critical point.

The present study unearthed three different CDW insulators induced by long-ranged interactions, however, some aspects need to be explored in the future. For instance, what happens if the hardcore approximation is relaxed? In this case, we anticipate that the new CDW insulators will emerge along with the MMI and SSMI states between every two trivial bosonic Mott insulators ($\rho_B = n$, with $n$ being an integer). This has already been observed for the MMI and SSMI phases, where the relations that they follow change to $\rho_B + \rho_F = n$ and $\rho_B + \rho_F^{\uparrow,(\downarrow)} = n$, respectively [68]. Either way, we do not discount the appearance of other insulators that could emerge from reducing the hardcore restriction.

This study is focused on a particular set of values for the local interspecies and intraspecies interactions, but it is important to think of the effect that changing these parameters can have on these insulators. By setting $U_{FF}$, the insulator states are expected to arise from critical values of the Bose–Fermi couplings, which will depend on each phase, as has been shown in previous studies without long-ranged interactions [66–70]. For instance, we observed that for $\rho_F = 4/5$, the stability of the SSCDW and ICDW insulators increases for higher values of the local interspecies interaction $U_{BF}$, since this parameter modulates the mutual behavior between species that characterize these CDW insulators. On the other hand, a small increase in the local fermionic interaction $U_{FF}$ can lead to a charge gap increase in the ICDW phase, due to the localization of the fermions in the CDW order, and a decrease in the SSCDW phase stability because of stronger perturbations caused by the free fermions on the insulator. However, a more thorough study has yet to be done.

The ground state evolution under the simultaneous effect of the next-neighbor interaction between fermions or bosons and/or the inclusion of interspecies long-ranged couplings was not widely explored. However, some preliminary results suggest that the insulator states unearthed in this paper will emerge. Moreover, we do not disregard the possibility of finding new states under these conditions. Nevertheless, this falls outside the scope of this work.

We trust that our investigation can inspire future research on the extended Bose–Fermi Hubbard model and specifically the previously shown incompressible phases since their CDW character presents an opportunity to observe new emerging behavior in mixtures that could give rise to possible applications in industry [100]. Fortunately, the ICDW phase has been observed in the excitonic system of GaAs bilayers recently [34], then an experimental observation of the other insulators would be a crucial input for the research on this topic. The latter could be done in the same condensed matter system where the long-ranged interactions of the dipolar excitons play a crucial role [33–37], while cold-atom setups also present a great opportunity, here the long-ranged interaction can be fulfilled using polar molecules [101, 102] which could be potentially loaded in an optical lattice, or Rydberg-dressed atoms [103–105] where proposals of extended Hubbard models have been made. These possible experimental realizations along with the present proposal of phases with long-ranged interactions open the pathway to an amalgam of new strongly-correlated insulators in low-dimensional systems.

## Acknowledgments

J. S.-V. thanks to the University of Pittsburgh for its kind hospitality during his sabbatical year. Gómez-Lozada thanks the Okinawa Institute of Science and Technology and the Quantum Systems Unit for their support and hospitality during his research internship.

**Funding information** Silva-Valencia acknowledges the support of the DIEB - Universidad Nacional de Colombia (Grant No. 51116).

## A  Particle-hole symmetry

In this appendix, we will show that the Hamiltonian (1)-(4) has a particle-hole symmetry in the hardcore limit for the bosons which allows us to simplify the phase diagrams in Fig. 1. First, we consider the symmetry of the fermions in the system and then we address the case for the bosons.

We introduce the charge conjugation operator $\hat{\mathcal{P}}$ which changes each filled Fock state for a vacuum one and vice-versa

$$\hat{\mathcal{P}}|1\rangle \to |0\rangle, \tag{A.1}$$

$$\hat{\mathcal{P}}|0\rangle \to |1\rangle. \tag{A.2}$$

Because of the Pauli exclusion principle, this operator is well-defined for each fermionic site. To generalize the concept for a set of fermionic sites the charge conjugation operator is defined through its effect on the creation and annihilation operators $\hat{f}^{\dagger}_{\sigma,j}$ and $\hat{f}_{\sigma,j}$, respectively [106]

$$\hat{\mathcal{P}}\hat{f}^{\dagger}_{\sigma,j}\hat{\mathcal{P}}^{-1} = (-1)^{j}\hat{f}_{\sigma,j}, \tag{A.3}$$

$$\hat{\mathcal{P}}\hat{f}_{\sigma,j}\hat{\mathcal{P}}^{-1} = (-1)^{j}\hat{f}^{\dagger}_{\sigma,j}, \tag{A.4}$$

for each spin $\sigma = \uparrow, \downarrow$ and site $j$. With these expressions, we see that the fermionic number operators $\hat{n}^{F}_{\sigma,j}$ transform under $\hat{\mathcal{P}}$ as

$$\hat{\mathcal{P}}\hat{n}^{F}_{\sigma,j}\hat{\mathcal{P}}^{-1} = (-1)^{2j}\hat{f}_{\sigma,j}\hat{f}^{\dagger}_{\sigma,j} \tag{A.5}$$

$$= 1 - \hat{n}^{F}_{\sigma,j}. \tag{A.6}$$

Now let us show that the fermionic Hamiltonian (3) is invariant under the effect of $\hat{\mathcal{P}}$. The fermionic hopping term shows directly this property

$$\hat{\mathcal{P}}\left[\hat{f}^{\dagger}_{\sigma,j}\hat{f}_{\sigma,j+1} + \text{H.c.}\right]\hat{\mathcal{P}}^{-1} = (-1)^{2j+1}\hat{f}_{\sigma,j}\hat{f}^{\dagger}_{\sigma,j+1} + \text{H.c.} \tag{A.7}$$

$$= \hat{f}^{\dagger}_{\sigma,j}\hat{f}_{\sigma,j+1} + \text{H.c.} \tag{A.8}$$

On the other hand, the fermionic on-site interaction term from (3) transforms as

$$\hat{\mathcal{P}}\left[\hat{n}^{F}_{\uparrow,j}\hat{n}^{F}_{\downarrow,j}\right]\hat{\mathcal{P}}^{-1} = \left(1 - \hat{n}^{F}_{\uparrow,j}\right)\left(1 - \hat{n}^{F}_{\downarrow,j}\right) \tag{A.9}$$

$$= \hat{n}^{F}_{\uparrow,j}\hat{n}^{F}_{\downarrow,j} + 1 - \left(\hat{n}^{F}_{\uparrow,j} + \hat{n}^{F}_{\downarrow,j}\right). \tag{A.10}$$

In this case, the interaction exchange is not directly invariant under the transformation. Nevertheless, the extra terms that appear are either constant or of the form of a chemical potential contribution, since we are working in a number-conserving Hilbert space these do not change the ground state of the Hamiltonian as their main effect is to shift the zero in the energy and chemical potential scales. For the fermionic next-neighbor interaction, we recover an analog result

$$\hat{\mathcal{P}}\left[\hat{n}^{F}_{j}\hat{n}^{F}_{j+1}\right]\hat{\mathcal{P}}^{-1} = \left(2 - \hat{n}^{F}_{j}\right)\left(2 - \hat{n}^{F}_{j+1}\right) \tag{A.11}$$

$$= \hat{n}^{F}_{j}\hat{n}^{F}_{j+1} + 4 - 2\left(\hat{n}^{F}_{j} + \hat{n}^{F}_{j+1}\right), \tag{A.12}$$

where the factors of 2 appear because $\hat{n}^{F}_{j} = \hat{n}^{F}_{\uparrow,j} + \hat{n}^{F}_{\downarrow,j}$.

For the bosons the charge conjugation operator (A.1) and (A.2) is ill-defined since there can be more than two states per site, nevertheless for the hardcore limit we extend the definition to the bosonic Hilbert space as

$$\hat{\mathcal{P}}\hat{b}^{\dagger}_{j}\hat{\mathcal{P}}^{-1} = \hat{b}_{j}, \tag{A.13}$$

$$\hat{\mathcal{P}}\hat{b}_{j}\hat{\mathcal{P}}^{-1} = \hat{b}^{\dagger}_{j}, \tag{A.14}$$

where we omit the $(-1)^{j}$ factor due to the bosonic nature of the many-body wave function. This does not affect the result from (A.5)-(A.6) since the hardcore bosons have the same on-site anti-commutation relations as the fermions, hence $\hat{\mathcal{P}}\hat{n}^{B}_{j}\hat{\mathcal{P}}^{-1} = 1 - \hat{n}^{B}_{j}$.

With (A.13)-(A.14) it is direct to show that

$$\hat{\mathcal{P}}\left[\hat{b}_j^\dagger \hat{b}_{j+1} + \text{H.c.}\right]\hat{\mathcal{P}}^{-1} = \hat{b}_j^\dagger \hat{b}_{j+1} + \text{H.c.} \tag{A.15}$$

Moreover, the general bosonic interaction transforms as

$$\hat{\mathcal{P}}\left[\hat{n}_j^{\text{B}}\hat{n}_l^{\text{B}}\right]\hat{\mathcal{P}}^{-1} = \hat{n}_j^{\text{B}}\hat{n}_l^{\text{B}} + 1 - \left(\hat{n}_j^{\text{B}} + \hat{n}_l^{\text{B}}\right). \tag{A.16}$$

Then, the bosonic Hamiltonian (2) is also invariant under this extended charge conjugation operator.

When we look at the interspecies Hamiltonian (4) it is noted that both definitions (A.3)-(A.4) and (A.13)-(A.14) together have the following effect on the interaction term

$$\hat{\mathcal{P}}\left[\hat{n}_j^{\text{B}}\hat{n}_l^{\text{F}}\right]\hat{\mathcal{P}}^{-1} = \hat{n}_j^{\text{B}}\hat{n}_l^{\text{F}} + 2 - \left(2\hat{n}_j^{\text{B}} + \hat{n}_l^{\text{F}}\right). \tag{A.17}$$

Hence, we conclude that (1)-(4) is invariant under the charge conjugation transformation up to constant and chemical potential terms. This means that if we find the corresponding ground state to a given set of parameters and densities of particles $\rho_{\text{B}}$, $\rho_{\text{F}}^{\uparrow}$ and $\rho_{\text{F}}^{\downarrow}$ because of the particle-hole symmetry this will also be the ground state for the fillings $1-\rho_{\text{B}}$, $1-\rho_{\text{F}}^{\uparrow}$ and $1-\rho_{\text{F}}^{\downarrow}$, respectively. The ground-state energies will differ because of the extra terms in the transformation of the interactions, but this will only apply a shift in both the energy scale and the chemical potential scale in the $\rho_{\text{B}} - \mu_{\text{B}}$ curves used along the present work, meaning that the physical properties of the system remain invariant. Because of this we only need to analyze half of the phase diagrams from Fig. 1.

# B Order of the phase transitions

In the main text, three different CDW insulators are characterized, and we suggest that future work should consider the study of the associated superfluid-insulator transition. Here, we provide a first insight into this topic by analyzing the system's ground-state energy as we increase the model's long-range interactions.

For each of the three fermionic densities from Sec. 3.1, 3.2, 3.3 we plot in Fig. 12 the ground state energy $E$ scaled by the system size and extrapolated to the thermodynamic limit as a function of $V_{\text{FF}} \in [0, 2]$ for each corresponding highlighted CDW insulator, which are the ICDW ($\rho_{\text{F}} = 2/3, \rho_{\text{B}} = 1/2$), MCDW ($\rho_{\text{F}} = 2/5, \rho_{\text{B}} = 2/5$) and the SSCDW ($\rho_{\text{F}} = 4/5, \rho_{\text{B}} = 2/5$), respectively. At $V_{\text{FF}} = 0$ the systems start in a superfluid phase, and as we increase $V_{\text{FF}}$ the superfluid-insulator transition occurs at a critical interaction showed as a vertical line of the corresponding color and line style. In all cases shown, the energy curve is smooth across the transition, from which we discard a first-order transition where a discontinuity in the energy's first derivative would generate a nudge in the curves. This agrees with previous literature on CDW phases which affirm its transition to be of second order [107]. Even then, a more meticulous study should be developed in the future for the present mixture system to characterize the nature of the phase transition.

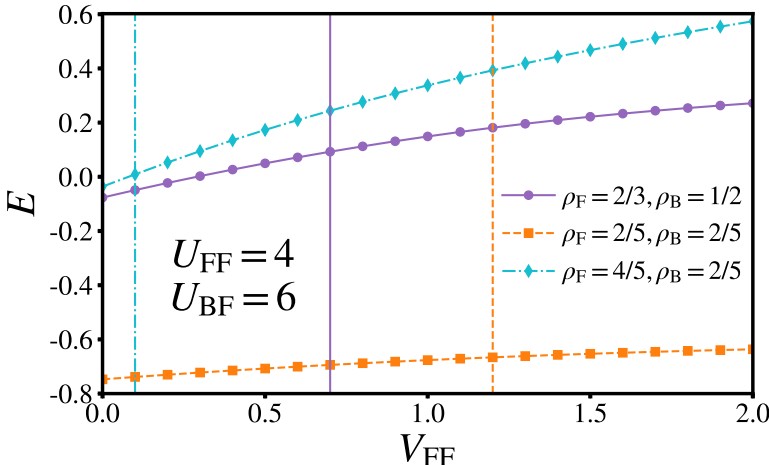

Figure 12: Ground state energy $E$ scaled by the system size in the thermodynamic limit as a function of $V_{FF}$. Three density combinations are shown, each associated with the emergence of a CDW insulator, corresponding to $\rho_F = 2/3$, $\rho_B = 1/2$ (purple, ICDW), $\rho_F = 2/5$, $\rho_B = 2/5$ (orange, MCDW) and $\rho_F = 4/5$, $\rho_B = 2/5$ (cyan, SSCDW), with onsite interactions $U_{FF} = 4$ and $U_{BF} = 6$. Vertical lines with the corresponding color and line style show the critical point of the transition for each insulating state, obtained in Sec. 3.1, 3.2, 3.3.

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
