# Peer review of "Insulator phases of Bose-Fermi mixtures induced by intraspecies next-neighbor interactions"

_SciPost Physics Core, doi:SciPost Phys. Core 8, 007 (2025)_

## Round 2 · Referee Report · Anonymous (Referee 1) · 2024-9-21

Strengths

1- identifying phases with insulating and density wave character in Bose-Fermi mixtures in one dimension 2- a rather large scan over parameter space in a model that has plenty of tuning knobs

Weaknesses

  • given the large local Hilbert space, the DMRG convergence is difficult
  • no discussion of phase transitions
  • no discussion of gapless phases; ie, a lot of physics is missing
  • as all parameters are in a regime of very large potential energy compared to kinetic energy, how much of the phase diagrams can be understood from mean-field or Gutzwiller types of approaches?
  • I find it hard to believe that no supersolids nor phases like CDW+superfluid are found.

Report

In "Insulator phases of Bose–Fermi mixtures induced by intraspecies next-neighbor interactions" the authors look at density wave structures in 1D Bose-Fermi hubbard models. They employ DMRG and provide several scans over bosonic chemical potentials at fixed fermionic densities. Most of the found density wave structures can be thought of small unit cells in which the bosonic and fermionic densities form commensurate combinations. The results are therefore not very surprising; in fact, the interesting physics in these models is usually found when tuning away from these near-classical configurations. In this sense I do not think that the paper meets the standards of a Scipost Physics paper. Either the authors can add something truly novel and unexpected, or they should consider a journal of substantially lower rank.

additional minor remarks: - given the hard-core nature of the bosons, what is their difference from the fermions up to a trivial Jordan-Wigner factor? I have not understood the differences between the presented results and the putative ones of the same system consisting of a 3-component hard-core bosonic system - on p4, when referring to cold atoms (refs 84-85), one should also include the scattering lengths as those cannot be chosen at will - in Fig 1, I am confused about the "white areas" mentioned in the caption. What is meant with that? I only see white areas and lines. - Fig 2b: is there a fermionic charge gap? - Fig 2b: why is this referred to as incommensurate? If we take 6 lattice sites, we will definitely find an integer number of particles. - Fig 6a, what is the small feature at rho_B = 0.2? - Fig 11b,c: what is the meaning of the tiny plateau at rho_b = 0.4?

Requested changes

On top of the comments made above I think the paper would benefit from a table summarizing the phases found and their respective order parameters

Recommendation

Accept in alternative Journal (see Report)

---

## Round 2 · Referee Report · Anonymous (Referee 2) · 2024-10-6

Strengths

  • Combination of two-species physics with more than onsite interactions.
  • Discussion of multitude of insulating phases.

Weaknesses

  • Motivation for multi-parameter model is vague.
  • The manuscript doesn't say much about the phase transitions (some order parameters are mentioned and critical values are extracted, yet what are the surrounding phases and the type of transition)

Report

In the manuscript by Gomez-Lozada et al studies quantum phases of a one-dimensional
model of Bose-Fermi mixtures with next-neighbor interspecies interactions.
The main interest is in insulating phases, as a main result, three incommensurate
insulators are discussed. These are the immiscible charge density wave state, the mixed
CDW state and the spin-selective CDW state. The study is carried out using the density
renormalization group technique and considers several filling fractions.

Bose-Fermi mixtures are one of the systems that are experimentally accessible
with quantum simulator plattforms while there are also a number of experimental
systems that yield extended Hubbard models or even long-range interactions.
There is also interest in CDW states or unusual isulating states, from a broad
range of angles. Thus this work certainly has a timely context.

However, as with all model studies that involve many parameters, one wonders about the
specific motivations for these multi-parameter models and the significance of the
results. Whenever there is one concrete experiment then certainly using very complex
models are justified, whenever entirely new physics emerges, the same holds. In the present
version, the motivation for this work is not spelled out in the most convincing way.

Certain other comments may warrant modifications of the manuscript, see the detailed list.
Overall, the manuscript may be publishable provided some improvements are implemented.

Requested changes

1- Improve motivation for multi-parameter model. 2- Terminology: Merely adding nearest-neighbor interactions does not justify the tem "non-local interactions" in my understanding. Please attempt a more precise definition. 3- Note that CDW states with periodicities larger than two are well-established in quantum magnetism ("magnetization plateaux") and multi-orbital models. Perhaps these analogies could be mentioned. 4- It is not entirely clear how the numerical accuracy is controlled. What is actually done? Is the maximum truncation error crucial or the sweep-dependent bond dimension? Please also provide definitions of the entropy/energy error. 5- Regarding Fig. 10, it is not entirely clear whether this is a schematic plot or obtained from numerical data. Also in Fig. 1: why are there only straight lines?

Recommendation

Ask for minor revision

---

## Round 4 · Referee Report · Anonymous (Referee 1) · 2024-12-6

Report

Compared to my previous report the authors seem to have addressed some issues, but not all. I will give one example, namely the nature of the phase transitions which the authors decided to address in the following way: The authors add a sentence in the text, referring to Appendix B and adding that it falls outside the scope in this paper. In Ref B there is an argument about a possible second order phase transition, and citing a review.
However, this question could be addressed more accurately, by checking if U(1) and lattice symmetries are simultaneously broken or not, and this in turn relates to another question about possible supersolid-like phases. The authors seem to have missed that.
For me, the reply is therefore not sufficient. Whether the paper meets the requirements of the journal is a question I leave to the editors with reference to my earlier report.

Recommendation

Accept in alternative Journal (see Report)

---

## Round 4 · Author Response

Dear Editor

Thank you for sending us the referees' comments on our paper entitled ``Insulator phases of Bose-Fermi mixtures induced by intraspecies next-neighbor interactions''.

We appreciate your advice, and a new version of the paper has been submitted to SciPost Physics Core. Furthermore, we are pleased to learn of the positive comments of the referees, and that both recommend the publication of the paper. In particular, the first referee states Accept in alternative Journal", and the second referee reports thatOverall, the manuscript may be publishable provided some improvements are implemented". Taking into account the suggestions raised by the referees, we have modified the paper as follows:

(i) The main criticism of both referees concerns the absence of discussion about the quantum phase transitions of the new phases reported in the paper. To resolve this issue, we have included an appendix that deals with the order of the phase transitions, where we have included a new figure of the ground-state energy versus the next-neighbor interactions, showing that all the transitions are continuous.

(ii) We have expanded the motivation of the multi-parameter model.

(iii) We have removed the term “incommensurable” and substituted “non-local” for “long-ranged”, since it's more appropriate for next-neighbor interactions.

(iv) We have improved the clarification of Fig. 1.

(v) We have added a further explanation of the numerical methods used.

(vi) We have added a explanation to Fig. 10.

(vii) We have added comments about the appearance of fermionic gaps in the presented mixed insulators.

(viii) We have included the following new references:

  • G. Gruner, Reviews of Modern Physics \textbf{60}, 1129 (1988)

  • A. van Otterlo and K.-H. Wagenblast, Physical Review Letters \textbf{72}, 3598 (1994)

  • G. G. Batrouni, R. T. Scalettar, G. T. Zimanyi and A. P. Kampf, Physical Review Letters \textbf{74}, 2527 (1995).

  • A. V. Chubukov and D. I. Golosov, Journal of Physics: Condensed Matter \textbf{3}, 69 (1991)

  • M. Oshikawa, M. Yamanaka and I. Affleck, Physical Review Letters \textbf{78}, 1984 (1997)

  • K. Totsuka, Physical Review B \textbf{57}, 3454 (1998).

  • A. A. Balandin, S. V. Zaitsev-Zotov and G. Grüner, Applied Physics Letters \textbf{119}, 170401 (2021).

We hope that with these modifications, which clarify and improve the presentation of the paper, the manuscript is appropriate for publication in SciPost Physics Core in its revised form.

Yours sincerely,

The authors.

---

## Round 4 · List of Changes

(i) The main criticism of both referees concerns the absence of discussion about the quantum phase transitions of the new phases reported in the paper. To resolve this issue, we have included an appendix that deals with the order of the phase transitions, where we have included a new figure of the ground-state energy versus the next-neighbor interactions, showing that all the transitions are continuous.

(ii) We have expanded the motivation of the multi-parameter model.

(iii) We have removed the term “incommensurable” and substituted “non-local” for “long-ranged”, since it's more appropriate for next-neighbor interactions.

(iv) We have improved the clarification of Fig. 1.

(v) We have added a further explanation of the numerical methods used.

(vi) We have added a explanation to Fig. 10.

(vii) We have added comments about the appearance of fermionic gaps in the presented mixed insulators.

(viii) We have included the following new references:

  • G. Gruner, Reviews of Modern Physics \textbf{60}, 1129 (1988)

  • A. van Otterlo and K.-H. Wagenblast, Physical Review Letters \textbf{72}, 3598 (1994)

  • G. G. Batrouni, R. T. Scalettar, G. T. Zimanyi and A. P. Kampf, Physical Review Letters \textbf{74}, 2527 (1995).

  • A. V. Chubukov and D. I. Golosov, Journal of Physics: Condensed Matter \textbf{3}, 69 (1991)

  • M. Oshikawa, M. Yamanaka and I. Affleck, Physical Review Letters \textbf{78}, 1984 (1997)

  • K. Totsuka, Physical Review B \textbf{57}, 3454 (1998).

  • A. A. Balandin, S. V. Zaitsev-Zotov and G. Grüner, Applied Physics Letters \textbf{119}, 170401 (2021).

---

## Editorial Decision

published